

# Regional infectious risk prediction of COVID-19 based on geo-spatial data

Xuewei Cheng[1], Zhaozhou Han[2], Badamasi Abba[1] and Hong Wang[1]

[1] School of Mathematics and Statistics, Central South University, China, Changsha, Hunan, China
[2] School of Economics, Jinan University, Guangzhou, Guangdong, China

## ABSTRACT

After the first confirmed case of the novel coronavirus disease (COVID-19) was found, it is of considerable significance to divide the risk levels of various provinces or provincial municipalities in Mainland China and predict the spatial distribution characteristics of infectious diseases. In this paper, we predict the epidemic risk of each province based on geographical proximity information, spatial inverse distance information, economic distance and Baidu migration index. A simulation study revealed that the information based on geographical economy matrix and migration index could well predict the spatial spread of the epidemic. The results reveal that the accuracy rate of the prediction is over 87.10% with a rank difference of 3.1. The results based on prior information will guide government agencies and medical and health institutions to implement responses to major public health emergencies when facing the epidemic situation.

## INTRODUCTION

The ongoing outbreak of coronavirus disease in 2019 (COVID-19) has caused 4,653 deaths, along with 86,045 confirmed cases and four suspected cases in China, as of June 18, 2020 (24:00 GMT+8), according to the National Health Commission of the People's Republic of China (NHCPRC, 2020, http://www.nhc.gov.cn/). The recorded deaths associated with COVID-19 notably exceeds the other two coronaviruses (severe acute respiratory syndrome coronavirus, SARS-CoV, and Middle East respiratory syndrome coronavirus, MERS-CoV). As the epidemic continues to spread, it poses a high threat to global public health and economy (*Bogoch et al., 2020*; *Wu, Leung & Leung, 2020*).

While the epidemic situation in Mainland China has been controlled, the outbreak outside Mainland China is reported to begin on a large scale. According to the World Health Organization (WHO), as of June 18 (2020), a total of 200 countries (or regions) have found confirmed cases, with a total of 14,406,440 confirmed cases and 601,846 deaths. The United States, India, Brazil, South Africa and Colombia are the top five countries with a more severe epidemic, among which 3,833,271 cases have been confirmed in The United States, with a total death of 142,877 people.

The emergence of COVID-19 coincides with the largest population migration season in China, that is, the spring festival tourism season. The virus spreads rapidly throughout the country. At the early stage of the outbreak, most cases were scattered, and some were

Corresponding author
Hong Wang, wh@csu.edu.cn

linked to the Huanan Seafood Wholesale Market (*Wu, Leung & Leung, 2020*). The Chinese government has implemented control measures, including setting up special hospitals and travel restrictions to mitigate the spread of the virus. Besides, the 31 provinces, districts and cities in Mainland China have also launched the first-level response to public health emergencies. On January 23, 2020, the local government of Wuhan suspended all public transport and closed all entry-exit traffic. Other cities in Hubei province announced similar traffic control measures shortly after Wuhan's instructions (*Lin et al., 2020*). Two months later, Wuhan, Hubei province, the thoroughfare of nine provinces, was reopened and "reconnected" with the outside world on April 8, 2020 (for specific major events, see Fig. 1-COVID-2019 timeline).

Since January 2020, many scholars have studied different aspects of the novel coronavirus, including biological characterization of the virus (*Lu et al., 2020*), medical diagnosis (*Liu et al., 2020*; *Lai et al., 2020*), clinical characteristics of patients (*Guan et al., 2020*; *Yang et al., 2020*; *Chen et al., 2020a*; *Shi et al., 2020*; *Chan et al., 2020*; *Chen et al., 2020b*), comparison with SARS (*Wilder-Smith, Chiew & Lee, 2020*), estimation of the reproductive number (*Zhang et al., 2020*; *Read et al., 2020*), future trends and the reporting ratio (*Liu et al., 2020*; *Benvenuto et al., 2020*; *Chen & Bai, 2020*). Some scholars use machine learning method to analyze the diagnosis and detection of COVID-2019 (*Hassantabar, Ahmadi & Sharifi, 2020*), gene expression programing and sensitivity analysis (*Dorosti et al., 2019*), forecast the economic growth (*Ahmadi et al., 2019*). Unlike other diseases, the prevention and suppression of transmission of infectious diseases have become particularly momentous.

After the first case is confirmed, scientific precautionary measures need to be taken. Due to the vast land of Mainland China and natural resources, the natural and economic conditions of each province and city are different. Thus, at the initial stage of the epidemic development, we need to estimate the risk of the epidemic in each province and provincial municipality, which could guide the authorities concern in carrying out preventive measures accordingly. Therefore, it is essential to forecast the development of the epidemic situation in each province and city, which will be related to the allocation of medical resources and ensure the supply of food, such as rice and water, to cater people's need.

*Dey et al. (2020)* use a visual exploratory data analysis approach to analyze the epidemiological outbreak of COVID-19. The result shows that it is highly momentous to readily provide information to begin the evaluation necessary to understand the risks and begin containment activities. *Stoecklin et al. (2020)* define a contact and follow-up procedure by the level of risk of infection and suggest that effective collaboration between all parties involved in the surveillance and response to emerging threats is required to detect imported cases early and to implement adequate control measures. *Kamel Boulos & Geraghty (2020)* offers pointers to, and describes, a range of practical online/mobile GIS and mapping dashboards and applications for tracking the 2019/2020 coronavirus epidemic and associated events as they unfold around the world. *Boldog et al. (2020)* developed a computing tool to assess the risk of outbreaks of COVID-2019 outside China, and consider key parameters, such as: (I) the evolution of cumulative number of cases

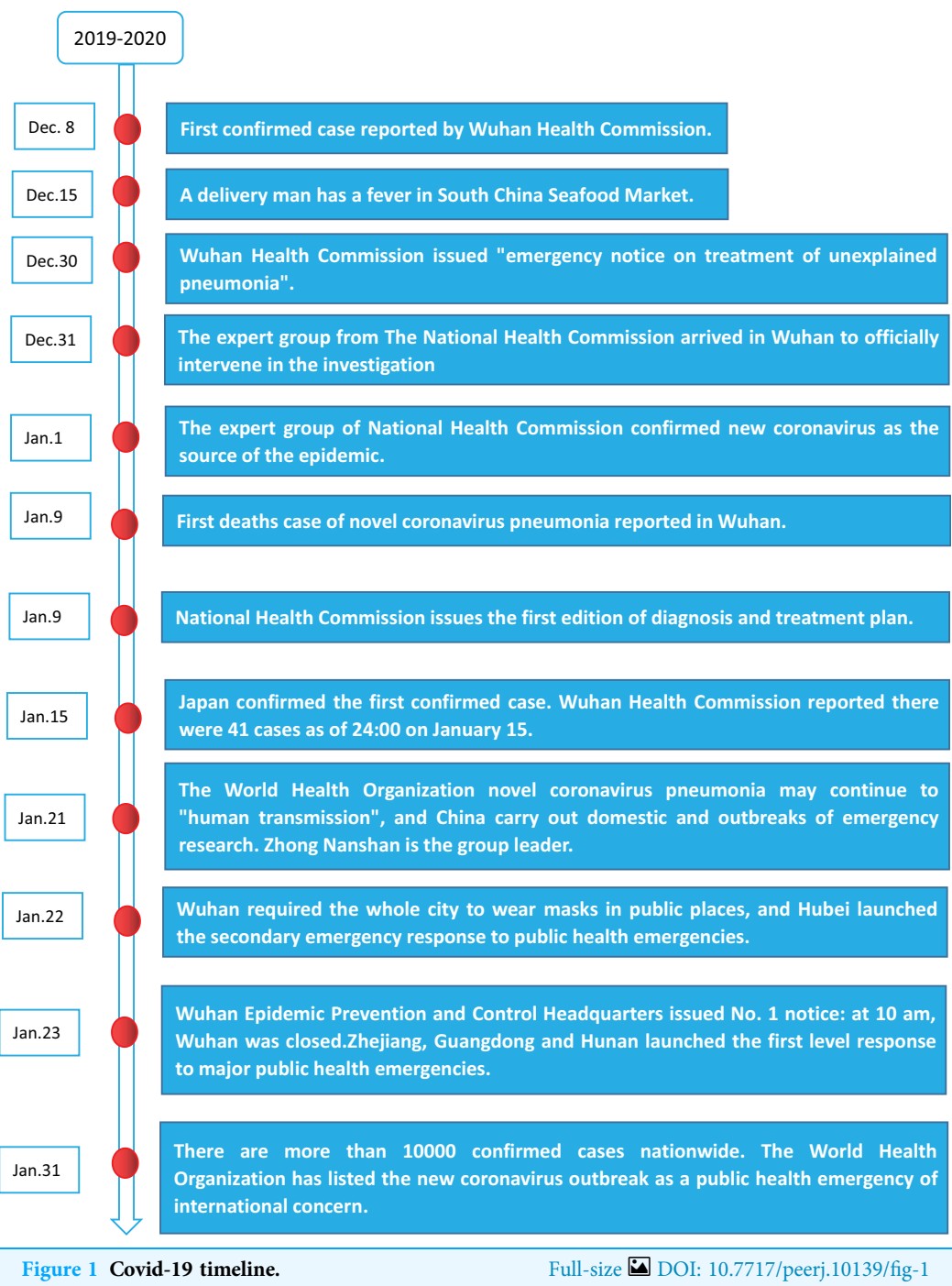

**Figure 1 Covid-19 timeline.**

outside mainland China; (II) connectivity between destination countries and China, including baseline travel frequency, travel restrictions and the effectiveness of entry inspection; (III) effectiveness of control measures in the country of destination. Many scholars assess the risks of other diseases or infectious diseases, such as cardiovascular (*Vusirikala et al., 2019*), CHIKV (*Mariconti et al., 2019*), H7N9 (*Zhang, Wang & Atkinson, 2019*), swine fever (*Mintiens et al., 2003*), severe dengue in Thailand (*Xu et al., 2019*).

*Ahmadi et al. (2020)* thought that the effect of climatic factors on spreading of COVID-2019 could play a vital role in the next coronavirus outbreak and the result of sensitivity analysis shows a direct relationship between the population density, intra-provincial movement and the infection outbreak.

The review of the novel coronavirus has made many outstanding achievements which relate to lots of aspects from the biological characteristics of the virus, the clinical characteristics of patients and the prediction of the number of people infected, future preventive measures as well as risk assessment. However, there are no much works in the literature on the division of regional risks in Mainland China. The existing literature only evaluated the epidemic situation and the related factors affecting the epidemic situation, and did not analyze the specific situation of regional epidemic situation, nor did it analyze how the regional differences of epidemic situation formed. These modeling methods are not specific to the region, which is not enough to provide accurate and useful control suggestions. Based on the researches before the event and little factual information available, the local governments can acquire some practical virus control advice to guide strategies for situational awareness and intervention, reducing the probability of regional infection, and suppressing the epidemic as soon as possible.

The development of spatial information systems and spatial econometrics has made it more efficient and convenient for us to measure the distance between regions (*Wieczorek & Hanson, 1997*; *Arslan, Cepni & Etiler, 2013*). *Boyda et al. (2019)* using GIS and spatial analysis methods, this paper reviewed and summarized the transmission of HIV in Africa. Due to the limitations of simple geographic information, *Kurata & Bapat (2015)* began to use the Euclidean distance matrix to measure the distance between two regions and generally used the Euclidean inverse distance to measure the proximity of two regions (*Zubaedah et al., 2020*; *Lumijarvi, Laurikkala & Juhola, 2004*). However, European inverse distance still cannot accurately reflect the strength of the relationship between the two regions, and hence, some economists have introduced economic distance into the spatial information system (*Bastawrous & Suni, 2020*; *Najafi Alamdarlo, 2018*). In the real epidemic prevention and control, population migration and motion can spread the epidemic rapidly, so it is an essential factor in the epidemic situation (*Wong et al., 2020*; *Yue & Clayton, 2011*). In this article, we introduced these factors into the spatial risk map theory to measure the distance between the regions (*Pourghasemi et al., 2020*; *Ijumulana et al., 2020*; *Fusade-Boyer et al., 2020*) and dynamically measure the spread of the epidemic in China. *Pourghasemi et al. (2020)* used a spatial model and risk map to analyze the epidemic situation in Iran from January 19, 2020, to July 14, 2020. The results show that the spatial model and risk map are significant for the analysis and simulation of epidemic spread.

In this article, we predict the spread trend of the epidemic, divide the risk level of provinces or provincial municipalities, to guide the governments of provinces or provincial municipalities to start the corresponding public health event response. It is achieved based on the initial location of the epidemic and according to the corresponding geographical distance, economic distance and Baidu migration index.

The main contribution of this paper is to use fewer epidemic data or even zero epidemic data to simulate the spread of an epidemic in Mainland China for a period of time. It is because we cannot get the data of the epidemic before it comes. Additionally, using the data of the epidemic situation itself to predict the epidemic situation, its practicality and generalization are not very strong. For better control the epidemic situation, it is necessary to divide the risk of the epidemic situation in each region, which is neither wholly sealed nor free control. However according to the risk level of the epidemic situation in different regions, the corresponding epidemic control measures should be taken. In the classification of the epidemic situation in 31 provinces or provincial municipalities in Mainland China, our method only predicted the epidemic level of four provinces or provincial municipalities incorrectly, which is an excellent academic achievement in the actual epidemic prevention and control.

## INSPIRATION AND NOVELTY

The first case of novel coronavirus confirmed in Wuhan was reported in December 2019. The study covers the epidemic situation in 31 provinces or provincial municipalities in Mainland China. Firstly, the most apparent feature of the epidemic development is that the number of confirmed cases in neighboring provinces or provincial municipalities of Wuhan is significantly higher than that in other parts of Mainland China. Thus, the authors verified whether the spread of the epidemic is related to the proximity of geographical space; secondly, the more developed cities are, the more infected cases recorded compared with economically backward provinces or provincial municipalities. Because the highly developed cities have closer economic exchanges with other cities, which resulted in a large number of population flows and hence the large-scale spread of the epidemic; thirdly, the migration data of the population will effectively assist the prediction of infectious diseases, especially the population emigration rate from the severely affected areas of the epidemic, which has immeasurable value for the prediction of infectious cases of the epidemic. Based on these three characteristics of the outbreaks, inspired by the actual epidemic data, this article studies the regional characteristics of the epidemic development, and predicts the provinces or provincial municipalities risk level of the epidemic.

As for the novelty and importance of this work, this research work is rarely seen in previous studies. First of all, as far as novelty is concerned, we try to use as little or no real data as possible to predict the real development of the epidemic situation, which is the starting point of our research work. Because using the data of the epidemic situation itself to predict the development of the epidemic in the future might be inappropriate. Because of the coming of the next large-scale epidemic, we do not have data on the epidemic itself, but we still need to carry out prevention and control of the epidemic. Zero epidemic data is the value of our research work, but can produce immeasurable value in the next outbreak.

The importance of this work is self-evident as the correctness of epidemic prevention and control not only involves the safety of life and property but also involves the stable operation of society. The prevention and control of the epidemic situation do not mean

that it is completely sealed off. Still, some put and lose, since total closure means economic stagnation, which is not an optimal strategy. We need to carry out corresponding control strategies according to the severity of the epidemic, which is the optimal trade-off between economic stagnation and epidemic prevention and control. This work is the foothold of our whole research, helping each region to classify the epidemic situation according to its severity.

## MATERIALS AND METHODS

### Description of the latest epidemic data

China had 86045 COVID-19 cases by 24:00 July 18, 2020, of which 2007 cases were imported from abroad. This article only studies the epidemic situation in China, so the imported cases are removed from the samples to obtain the domestic epidemic data. The specific epidemic data and ranking are shown in Table 1.

At present, the epidemic situation in Mainland China is stable, with 68,135 confirmed cases. Hubei Province, the center of the epidemic situation, account for 79.19% of the total confirmed cases. It shows that there is a strong relationship between the spread of the epidemic and the regional spread speed of the epidemic is very fast. In the six provinces adjacent to Hubei, namely Anhui, Hunan, Shaanxi, Jiangxi, Henan and Chongqing, the number of confirmed cases reached 5,123, accounting for 5.95% of the total confirmed cases. It shows that in the early stage of the epidemic, as long as the seven provinces are well controlled, 85.14% of the cases in the whole country can be stabilized, which is an essential measure for epidemic prevention and control. With the global spread of the epidemic, China's imported cases began to increase, especially in economically developed provinces, such as Beijing, Shanghai, Guangdong; and some import and export and tourism provinces, such as Heilongjiang and Inner Mongolia.

### Epidemic prediction based on spatial geographic adjacency information

To study the spatial clustering characteristics of the novel coronavirus disease, we first examine the geographical spatial correlation between regions. The spatial weight $W$ adopts a simple geographical weight, that is, for 31 provinces or provincial municipalities across the country, the weight of 1 is assigned if there is a common boundary between them, 0 and if they are not adjacent. We simply call this kind of spatial matrix as a 0–1 matrix, and its basic form is

$$w_{ij} = \begin{cases} 1 & i \text{ and } j \text{ are adjacent} \\ 0 & i \text{ and } j \text{ are not adjacent} \end{cases} (i \neq j) \tag{1}$$

Table 2 displays the neighboring information of 31 provinces or provincial municipalities in Mainland China.

According to the response of major public health emergencies in China, there are four levels (I, II, III and IV). Therefore, it is necessary to divide the epidemic situation into four levels based on the risk and severity, to facilitate the corresponding provinces or provincial municipalities to initiate the corresponding health emergencies response. Based

**Table 1 Domestic epidemic situation in China (as of 18 June 2020, 24:00).**

| Region | Cumulative confirmed | Overseas import | Domestic confirmed | Epidemic level | Rank |
|---|---|---|---|---|---|
| Hubei | 68,135 | 1 | 68,134 | 1 | 1 |
| Guangdong | 1,659 | 264 | 1,395 | 2 | 2 |
| Henan | 1,276 | 3 | 1,273 | 2 | 3 |
| Zhejiang | 1,270 | 51 | 1,219 | 2 | 4 |
| Hunan | 1,019 | 1 | 1,018 | 2 | 5 |
| Anhui | 991 | 1 | 990 | 2 | 6 |
| Jiangxi | 932 | 3 | 929 | 2 | 7 |
| Shandong | 797 | 34 | 763 | 3 | 8 |
| Beijing | 929 | 174 | 755 | 3 | 9 |
| Jiangsu | 655 | 24 | 631 | 3 | 10 |
| Sichuan | 599 | 4 | 595 | 3 | 11 |
| Chongqing | 583 | 7 | 576 | 3 | 12 |
| Heilongjiang | 947 | 386 | 561 | 3 | 13 |
| Shanghai | 733 | 391 | 342 | 3 | 14 |
| Hebei | 349 | 10 | 339 | 3 | 15 |
| Fujian | 364 | 68 | 296 | 3 | 16 |
| Guangxi | 255 | 3 | 252 | 3 | 17 |
| Shaanxi | 322 | 79 | 243 | 3 | 18 |
| Yunnan | 188 | 14 | 174 | 3 | 19 |
| Hainan | 171 | 2 | 169 | 3 | 20 |
| Jilin | 155 | 5 | 150 | 4 | 21 |
| Guizhou | 147 | 1 | 146 | 4 | 22 |
| Tianjin | 203 | 66 | 137 | 4 | 23 |
| Shanxi | 201 | 67 | 134 | 4 | 24 |
| Liaoning | 164 | 33 | 131 | 4 | 25 |
| Gansu | 167 | 75 | 92 | 4 | 26 |
| Xinjiang | 106 | 14 | 92 | 4 | 27 |
| Inner Mongolia | 249 | 172 | 77 | 4 | 28 |
| Ningxia | 75 | 2 | 73 | 4 | 29 |
| Qinghai | 18 | 1 | 17 | 4 | 30 |
| Tibet | 1 | 0 | 1 | 4 | 31 |

**Notes:**
Data source: Sina News real-time dynamic tracking of novel coronavirus disease (all data sources in this article are from Sina News, if not specified).
Data link: https://news.sina.cn/zt_d/yiqing0121?ua=iPhone10%2C2__weibo__10.1.1__iphone__os12.4.1&from=10A1193010&wm=3049_0135.

on the above geographical adjacency information (GAI), we partitioned the 31 provinces or provincial municipalities into four risk levels with Hubei as the center of infectious diseases. Table 3 presents the specific hazard classification. According to the geographical proximity between the province and the epidemic center, the highest risk level is level 1; if the province is adjacent to the epidemic center, it is level 2; if the province is adjacent to a level 2 Province, it is level 3, and so on. To show the transmission process of

**Table 2 The adjacent information of 31 provinces or provincial municipalities.**

| S/N | Region | Adjacent information | S/N | Region | Adjacent information |
|---|---|---|---|---|---|
| 1 | Beijing | 2, 3 | 17 | Hubei | 12, 14, 16, 18, 22, 27 |
| 2 | Tianjin | 1, 3 | 18 | Hunan | 14, 17, 19, 20, 22, 24 |
| 3 | Hebei | 1, 2, 4, 5, 6, 15, 16 | 19 | Guangdong | 13, 14, 18, 20, 21 |
| 4 | Shanxi | 3, 5, 16, 27 | 20 | Guangxi | 18, 19, 24, 25 |
| 5 | Inner Mongolia | 3, 4, 6, 7, 8, 27, 28, 30 | 21 | Hainan | 19 |
| 6 | Liaoning | 3, 5, 7 | 22 | Chongqing | 17, 18, 23, 24, 27 |
| 7 | Jilin | 5, 6, 8 | 23 | Sichuan | 22, 24, 25, 26, 27, 28, 29 |
| 8 | Heilongjiang | 5, 7 | 24 | Guizhou | 18, 20, 22, 23, 25 |
| 9 | Shanghai | 10, 11 | 25 | Yunnan | 20, 23, 24, 26 |
| 10 | Jiangsu | 9, 11, 12, 15 | 26 | Tibet | 23, 25, 29, 31 |
| 11 | Zhejiang | 9, 10, 12, 13, 14 | 27 | Shaanxi | 4, 5, 16, 17, 22, 23, 28, 30 |
| 12 | Anhui | 10, 11, 14, 15, 16, 17 | 28 | Gansu | 5, 23, 27, 29, 30, 31 |
| 13 | Fujian | 11, 14, 19 | 29 | Qinghai | 23, 26, 28, 31 |
| 14 | Jiangxi | 11, 12, 13, 17, 18, 19 | 30 | Ningxia | 5, 27, 28 |
| 15 | Shandong | 3, 10, 12, 16 | 31 | Xinjiang | 26, 28, 29 |
| 16 | Henan | 3, 4, 12, 15, 17, 27 | | | |

**Note:**
The numbers of provinces or provincial municipalities in this paper are in the order of Table 2.

**Table 3 Hazard classification of provinces or provincial municipalities based on geographical proximity information.**

| Hazard level | S/N of provinces or provincial municipalities |
|---|---|
| Level one | 17 |
| Level two | 12, 14, 16, 18, 22, 27 |
| Level three | 3, 4, 5, 10, 11, 13, 15, 19, 20, 23, 24, 28, 30 |
| Level four | 1, 2, 6, 7, 8, 9, 21, 25, 26, 29, 31 |

the epidemic more clearly, we have drawn Fig. 2. The case first spread in the center of the epidemic 17 (Hubei), then to its neighboring provinces (1–2 level), then to its neighboring provinces too (2–3 level), and finally to 31 provinces or provincial municipalities in China (3–4 level).

From Table 3 and Fig. 2, we concluded that with Hubei as the center, all other provinces or provincial municipalities in Mainland China are reachable through at most two provinces or provincial municipalities. As the capital of Hubei Province, Wuhan is known as the "thoroughfare of nine provinces". It has the largest water, land and air transportation hub in endoland region of China and the shipping center in the middle reaches of the Yangtze River. Its high-speed rail network connects most part of the country, and it is the only city in the central part of China that can directly navigate five of the World continents. Therefore, the geographical location of Wuhan, Hubei Province, led to the national outbreak of the novel coronavirus disease.
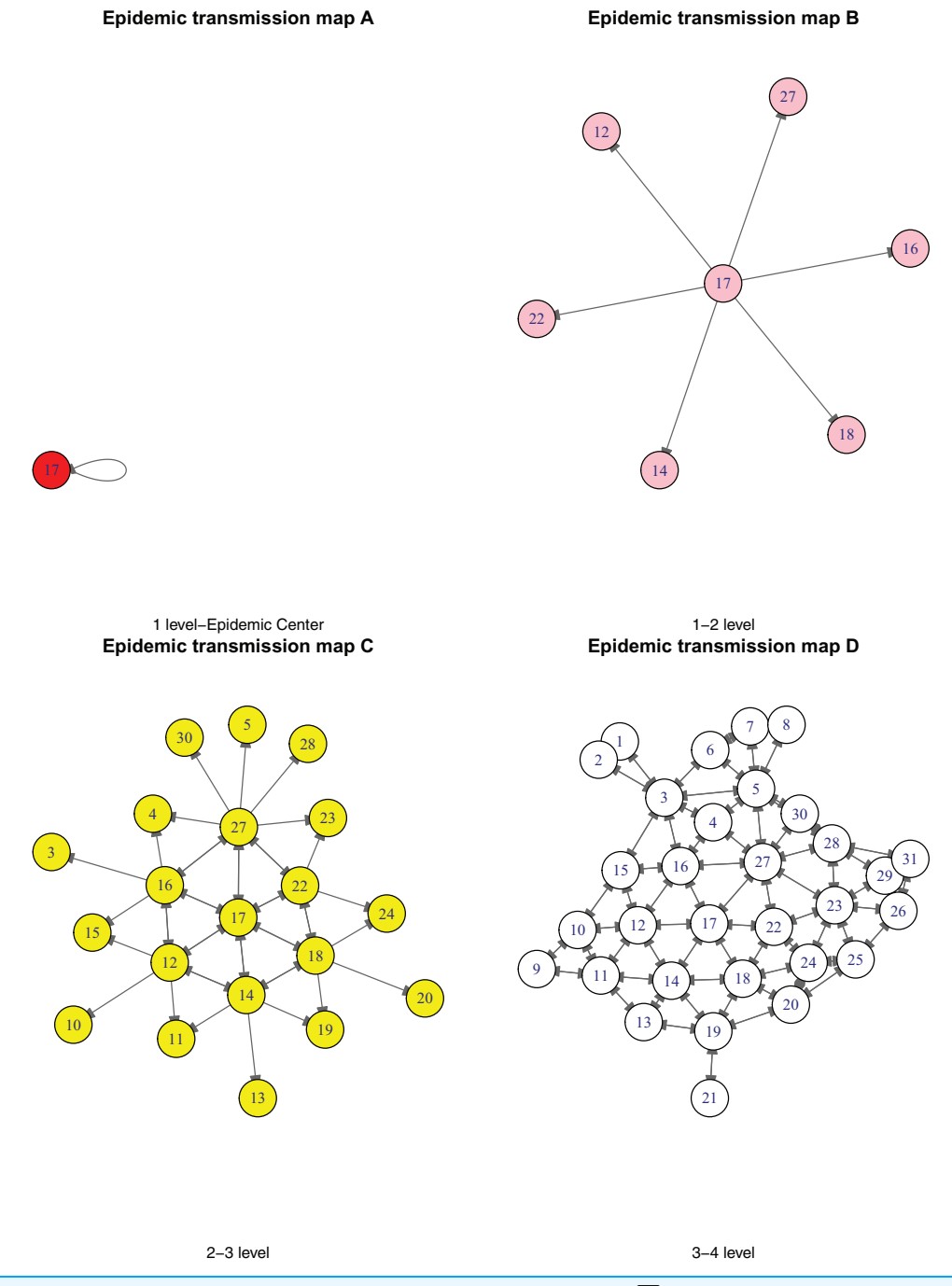

**Figure 2  Epidemic transmission map (A–D).**

However, if only based on the 0–1 adjacency of geographic information, the interaction infection between provinces or provincial municipalities is not considered. For example, the epidemic situation in Hubei may affect Anhui, Jiangxi, Henan, Hunan, Chongqing and Shaanxi. In turn, the epidemic situation in Anhui may also affect Hubei, Jiangsu, Jiangxi, Shandong, Henan and Zhejiang. In consideration of this mutual influence, we still take Hubei as the center of the epidemic situation, assigning Hubei "epidemic index 1"

**Table 4 Epidemic index and ranking of 31 provinces or provincial municipalities in China.**

| Region | Epidemic index | Rank | Region | Epidemic index | Rank |
|---|---|---|---|---|---|
| Hubei | 1.272 | 1 | Shanxi | 0.121 | 17 |
| Anhui | 0.330 | 2 | Guangxi | 0.118 | 18 |
| Shaanxi | 0.328 | 3 | Fujian | 0.113 | 19 |
| Jiangxi | 0.326 | 4 | Tibet | 0.090 | 20 |
| Hunan | 0.315 | 5 | Ningxia | 0.077 | 21 |
| Henan | 0.298 | 6 | Qinghai | 0.075 | 22 |
| Chongqing | 0.271 | 7 | Xinjiang | 0.071 | 23 |
| Cichuan | 0.205 | 8 | Jilin | 0.070 | 24 |
| Zhejiang | 0.191 | 9 | Liaoning | 0.064 | 25 |
| Inner Mongolia | 0.183 | 10 | Yunnan | 0.059 | 26 |
| Hebei | 0.177 | 11 | Shanghai | 0.049 | 27 |
| Jiangsu | 0.156 | 12 | Heilongjiang | 0.049 | 28 |
| Guangdong | 0.142 | 13 | Beijing | 0.047 | 29 |
| Guizhou | 0.136 | 14 | Tianjin | 0.040 | 30 |
| Gansu | 0.133 | 15 | Hainan | 0.031 | 31 |
| Shandong | 0.126 | 16 | | | |

**Table 5 Risk classification of provinces or provincial municipalities based on epidemic index.**

| Hazard level | Division results of epidemic index | Real division results |
|---|---|---|
| Level one | 17 | 17 |
| Level two | 12, 14, 16, 18, 22, 27 | **11**, 12, 14, 16, 18, **19** |
| Level three | 3, 4, 5, 10, 11, 13, 15, 19, 20, 23, 24, 26, 28 | **1**, 3, **8**, **9**, 10, 13, 15, 20, **21**, **22**, 23, **25**, **27** |
| Level four | 1, 2, 6, 7, 8, 9, 21, 25, 29, 30, 31 | 2, *4*, *5*, 6, 7, *24*, *26*, *28*, 29, 30, 31 |

Note:
The figures in bold and italics indicate that the predicted results are different from the real ones. To be specific, bold numbers mean that the prediction level is low, and the italics mean that the prediction level is high. The bold and italicized numbers in other tables have the same meaning as this table.

according to the principle of outward diffusion and one-step diffusion (*Martini, Nelson & Dahmus, 2014*), and spread its epidemic index outward. Six neighboring provinces or provincial municipalities obtained 1/6 of the epidemic index, respectively. By analogy, after Anhui obtained the epidemic index, it was spread to six neighboring provinces or provincial municipalities, each of which obtained 1/6 of the epidemic index of Anhui. After simple programing and simulation[1], the corresponding epidemic index and ranking are obtained as presented in Table 4.

Through simple geographical adjacency and epidemic transmission principle, the epidemic index can reflect the risk situation of provinces or provincial municipalities. Although there is a particular gap in the real situation, it has specific reference value for epidemic prevention. The risk level of provinces or provincial municipalities according to the epidemics index is given in Table 5, keeping the consistency of the number of provinces at each level reference to the risk level provides in Table 3.

[1] The principle of programming is that according to the theory that the epidemic situation spreads from the center to the outside, the epidemic index is equally distributed to all the neighbors connected with it. The provinces orprovincial municipalities that get the epidemic index are then transmitted to all their neighbors. After a round of transmission, we calculate the epidemic transmission index of 31 provinces and provincial municipalities.

[2] The real division results are based on the ranking in Tables 1 and 5.

In comparison with the real data[2], we found that there is a difference between the predicted and the real level of epidemic in the provinces or provincial municipalities and only depend on the geographical adjacency. From the epidemic index and risk level (Tables 4 and 5), 11 (Zhejiang) and 19 (Guangdong) of the third level risk provinces or provincial municipalities shifted down to the second level; 22 (Chongqing) and 27 (Shaanxi) of the second level risk provinces or provincial municipalities moved up to the third level risk provinces or provincial municipalities. This gap indicates that the spread of the epidemic is not only related to the geographical proximity, but also the economic level. The higher the economic level of a province is, the closer its interaction with other provinces is, which is more likely to lead to the spread of novel coronavirus disease. We can see the difference between the estimated results of the epidemic index of the third risk level provinces or provincial municipalities and the real risk level (Table 5-Real division results), which further verifies the previous hypothesis. 1 (Beijing) and 9 (Shanghai) entered the third level of the real situation (Table 5) from the fourth level of the epidemic index; 5 (Inner Mongolia), 24 (Guizhou), 26 (Tibet) and 28 (Gansu) entered the fourth level of the real situation from the third level of the epidemic index. To sum up, we need to find a matrix that can measure the economic distance between every two provinces and then apply it to explain the transmission of novel coronavirus disease.

## Epidemic prediction based on spatial Euclidean inverse distance

Before we carry out the impact of economic distance on the spread of the epidemic, we need to measure the impact of Euclidean distance on the spread of the epidemic. It is because, the simple spatial geographic information is adjacent to each other, whether it can comprehensively and accurately measure the spread of the epidemic or not remains a discussion issue. The point is that, can we verify whether the spread of epidemic will decline with farther distance, based on the spatial Euclidean inverse distance (EID)? Instead of geographical adjacency information, between all cities given the same weight index. Therefore, from the longitude and latitude data of the provincial capitals of each province in Mainland China, we determine the Euclidean distance $d_{ij}$ for each pair of provinces ($i$ and $j$), and then construct the space inverse distance matrix based on the Euclidean distance matrix, which is the reciprocal of the distance between space elements. Following the expression of the Euclidean distance matrix.

$$W_{ij} = \begin{cases} 1/d_{ij} & i \neq j \\ 0 & i = j \end{cases} \tag{2}$$

Space anti-Euclidean distance matrix can realize simple multi-level propagation. That is to say, novel coronavirus disease cases in one province or municipality may spread to 30 other provinces and municipalities throughout the country. The closer the distance, the higher the impact coefficient of the epidemic. Based on the theory of distance inverse weighting (Zubaedah et al., 2020), we still take Hubei as the center of the epidemic situation, according to the principle of outward diffusion and multi-step diffusion, we assign Hubei "epidemic index 1" and spread its epidemic index outward. The remaining

**Table 6 Epidemic index and ranking of 31 provinces or provincial municipalities in Mainland China.**

| Region | Epidemic index | Rank | Region | Epidemic Index | Rank |
|---|---|---|---|---|---|
| Hubei | 24.553 | 1 | Xinjiang | 2.378 | 17 |
| Hunan | 3.938 | 2 | Tibet | 2.334 | 18 |
| Jiangxi | 3.520 | 3 | Gansu | 2.334 | 19 |
| Anhui | 3.146 | 4 | Sichuan | 2.317 | 20 |
| Guangdong | 2.850 | 5 | Guizhou | 2.294 | 21 |
| Henan | 2.823 | 6 | Ningxia | 2.283 | 22 |
| Fujian | 2.799 | 7 | Heilongjiang | 2.262 | 23 |
| Jiangsu | 2.746 | 8 | Shanxi | 2.245 | 24 |
| Zhejiang | 2.678 | 9 | Hebei | 2.245 | 25 |
| Shanxi | 2.626 | 10 | Inner Mongolia | 2.238 | 26 |
| Shaanxi | 2.584 | 11 | Liaoning | 2.203 | 27 |
| Guangxi | 2.500 | 12 | Beijing | 2.178 | 28 |
| Chongqing | 2.484 | 13 | Qinghai | 2.105 | 29 |
| Shanghai | 2.405 | 14 | Jilin | 2.087 | 30 |
| Yunnan | 2.397 | 15 | Tianjin | 2.068 | 31 |
| Hainan | 2.379 | 16 | | | |

Note:
After the epidemic index is spread, it is standardized to make the total epidemic index of all provinces or provincial municipalities be 100.

30 provinces or provincial municipalities in the country obtained the epidemic index $1/d_{ij}$, respectively. By analogy, after Anhui obtained the epidemic index, then each of the remaining 30 provinces or the provincial municipalities in Anhui, obtained the epidemic index $1/d_{ij}$ of Anhui. In this way, the transmission of epidemic index in all provinces or provincial municipalities is completed, and then calculated the final epidemic index obtained in each province and city. After programming and simulation[3], Table 6 displays the corresponding epidemic index and ranking.

After the inverse weighting of Euclidean distance, the ranking of epidemic index becomes more reasonable. Therefore, according to the hazard classification in Table 3, the hazard levels of provinces or provincial municipalities based on the Euclidean inverse distance are shown in Table 7 below.

From Table 7, we can observe that the simulation results are more reasonable than that of Table 5. For instance, Guangdong (19) is not adjacent to Hubei Province, but it leaps from the third level dangerous city in Table 5, to the second level dangerous city under the inverse distance matrix, which is more consistent with the real situation. The number of errors in the three dangerous provinces or provincial municipalities of the predicted results and the real epidemic situation level has changed from 7 in Table 5 to 4 in Table 7, which is more accurate and reasonable. However, there are still some shortcomings. 11 (Zhejiang), 1 (Beijing) and 3 (Hebei) are far away from Hubei, but their predicted risk levels based on the inverse distance matrix all show a backward shift. Similarly, 26 (Tibet), 28 (Gansu) and 31 (Xinjiang) are economically backward provinces,

[3] The principle of programing is based on the theory of epidemic spreading from the center to the outside. However, this simulation is different from the simulation implementation in "Epidemic Prediction Based on Spatial Geographic Adjacency Information", because the spread from each region to another region is not completely equal weight spread, but the reciprocal weighted spread based on distance. In other words, the closer the two regions are, the greater the epidemic index will be. See code in the Supplemental Information for details.

**Table 7 Risk classification of provinces or provincial municipalities based on epidemic index.**

| Hazard level | Division results of epidemic index | Real division results |
|---|---|---|
| Level one | 17 | 17 |
| Level two | 12, 13, 14, 16, 18, 19 | 11, 12, 14, 16, 18, 19 |
| Level three | 9, 10, 11, 15, 20, 21, 22, 23, 25, 26, 27, 28, 31 | 1, 3, 8, 9, 10, 13, 15, 20, 21, 22, 23, 25, 27 |
| Level four | 1, 2, 3, 4, 5, 6, 7, 8, 24, 29, 30 | 2, 4, 5, 6, 7, 24, 26, 28, 29, 30, 31 |

but the predicted risk levels have moved forward. Based on the above analysis, it is necessary to introduce economic distance to simulate the spread of the epidemic.

## Epidemic prediction based on geographical distance and economic distance

Predicting the spread of the epidemic considering only the geographical distance of the provinces or provincial municipalities can reflect limited information. Therefore, we further take into account the economic influence among regions, to better reflect the reality of epidemic transmission. For example, Tianjin connects with Beijing and Hebei, but the two regions have different economic ties with Tianjin. Based on China's economic reality, we assume that the closer the economic exchanges between two cities with the same economic level, the higher the spread of the epidemic, and therefore this study uses the difference of per capita GDP between regions as an indicator to measure the economic distance between regions referring to *Lin, Long & Wu (2005)*. Its basic form is

$$E_{ij} = \begin{cases} \bar{y}_i - \bar{y}_j & i \neq j \\ 0 & i = j \end{cases} \tag{3}$$

where, $\bar{y}_i$ is the average per capita GDP of the $i$th province and city in 2014–2018. We construct the geographical-economic matrix (combining the geographical and economic information) based on the spatial inverse distance matrix and economic distance matrix (Standardization). The basic form of the geographical-economic matrix $W^*$ is

$$W_{ij}^* = W_{ij} + \lambda E_{ij} \tag{4}$$

where, $\lambda$ is the weighting coefficient of geographical distance and economic distance. A simulated result[4] for the epidemic index and its ranking, taken $\lambda = 0.5$, is presented in Table 8.

According to the hazard classification in Table 3, we present the hazard levels of provinces or provincial municipalities based on the geographical economic distance (GED) in Table 9.

Form Table 8, the result of the matrix of geographical-economic, indicates that it may be more reasonable to predict the spread of the epidemic, taking into account both the geographical and economic distances. The ranking of 1 (Beijing) increased from 28 to 23, and that of 2 (Tianjin) increased from 31 to 28, but the predicted risk level is still not well adjusted. 13 (Fujian) prediction results still move forward, because Fujian itself is relatively close to Hubei Province in geographical distance, and the per capita GDP is

[4] The principle of programing is based on the theory of epidemic spreading from the center to the outside. However, this simulation is different from the simulation implementation in "Epidemic Prediction Based on Spatial Geographic Adjacency Information" and "Epidemic Prediction Based on Spatial Euclidean Inverse Distance", because the spread from each region to another region is not completely equal weight spread, but the reciprocal weighted spread based on geographical and economic distance. In other words, the closer the two regions are in geographical and economic distance, the greater the epidemic index will be. See code in the Supplemental Information for details.

Table 8 Epidemic index and ranking of 31 provinces or provincial municipalities in Mainland China (Geographic economic matrix).

| Region | Epidemic Index | Rank | Region | Epidemic Index | Rank |
|---|---|---|---|---|---|
| Hebei | 27.488 | 1 | Hainan | 2.307 | 17 |
| Hunan | 3.464 | 2 | Guizhou | 2.306 | 18 |
| Jiangxi | 3.186 | 3 | Shandong | 2.291 | 19 |
| Anhui | 2.901 | 4 | Sichuan | 2.285 | 20 |
| Henan | 2.655 | 5 | Chongqing | 2.266 | 21 |
| Fujian | 2.627 | 6 | Shanxi | 2.265 | 22 |
| Guangdong | 2.627 | 7 | Beijing | 2.252 | 23 |
| Jiangsu | 2.625 | 8 | Heilongjiang | 2.245 | 24 |
| Zhejiang | 2.524 | 9 | Hebei | 2.234 | 25 |
| Guangxi | 2.500 | 10 | Ningxia | 2.204 | 26 |
| Shanghai | 2.439 | 11 | Inner Mongolia | 2.191 | 27 |
| Yunan | 2.390 | 12 | Tianjin | 2.183 | 28 |
| Shaanxi | 2.383 | 13 | Qinghai | 2.122 | 29 |
| Gansu | 2.337 | 14 | Liaoning | 2.061 | 30 |
| Tibet | 2.330 | 15 | Jilin | 1.995 | 31 |
| Xinjiang | 2.317 | 16 | | | |

relatively high. The epidemic index of this kind of provinces or provincial municipalities is challenging to measure simply by geographical distance and economic distance, so the next part of this paper will introduce Baidu's migration index.

## Epidemic prediction based on migration index weighting of Baidu

Usually, when an epidemic occurs, the more people moving out of a central city, the higher the spread of the epidemic. The spread of the epidemic can thus be closely related to the number of people moving out of the central city (Wuhan). Baidu[5] provides daily real-time migration rate; therefore, we quote January 28 and 29[6] and No. 30, the emigration rate from Wuhan to 31 provinces or provincial municipalities in Mainland China, and a simple average, as the migration index from Wuhan. In this paper, the epidemic index generated by geographical-economic distance and the Baidu migration index (BMI) is used to weigh the impact of the Hubei epidemic on the whole country. Simulation study[7] reveals that when the epidemic index and the Baidu migration index are 1:2 weighted, it well reflects the spread of the epidemic. Table 10 gives the weighted epidemic index.

After the weighting of the Baidu migration index, the predicted epidemic index and ranking are more closely in line with the reality of epidemic transmission. In order to understand the difference between the ranking of the epidemic index and the real situation, we took the rank of the ranked difference by using the real epidemic data on March 4, 2020, as shown in Table 11 below.

[5] Baidu migration index website: https://qianxi.baidu.com/? From = Shoubai × city = 0.

[6] On January 29, Wuhan issued the order to seal the city, so this paper selects the day before and the day after as the outward migration index of Wuhan.

[7] The principle of programing is based on the theory of epidemic spreading from the center to the outside. However, this simulation is different from the simulation implementation in "Epidemic Prediction Based on Spatial Geographic Adjacency Information" to "Epidemic Prediction Based on Geographical Distance" and economic distance, because the spread from each region to another region is not completely equal weight spread, but the reciprocal weighted spread based on geographical, economic distance and Baidu Index. See code in the Supplemental Information for details.

**Table 9  Risk classification of provinces or provincial municipalities based on epidemic index.**

| Hazard level | Division results of epidemic index | Real division results |
|---|---|---|
| Level one | 17 | 17 |
| Level two | 12, 13, 14, 16, 18, 19 | 11, 12, 14, 16, 18, 19 |
| Level three | 9, 10, 11, 15, 20, 21, 23, 24, 25, 26, 27, 28, 31 | 1, 3, 8, 9, 10, 13, 15, 20, 21, 22, 23, 25, 27 |
| Level four | 1, 2, 3, 4, 5, 6, 7, 8, 22, 29, 30 | 2, 4, 5, 6, 7, 24, 26, 28, 29, 30, 31 |

**Table 10  Epidemic index and ranking of 31 provinces or provincial municipalities in China (weighted by Baidu migration index).**

| Region | Epidemic index | Rank | Region | Epidemic index | Rank |
|---|---|---|---|---|---|
| Hubei | 69.303 | 1 | Hebei | 0.869 | 17 |
| Hunan | 2.119 | 2 | Beijing | 0.842 | 18 |
| Henan | 2.027 | 3 | Hainan | 0.836 | 19 |
| Guangdong | 1.707 | 4 | Guizhou | 0.827 | 20 |
| Jiangxi | 1.558 | 5 | Gansu | 0.824 | 21 |
| Anhui | 1.407 | 6 | Shanxi | 0.815 | 22 |
| Jiangsu | 1.322 | 7 | Xinjiang | 0.795 | 23 |
| Zhejiang | 1.081 | 8 | Heilongjiang | 0.788 | 24 |
| Fujian | 0.985 | 9 | Tibet | 0.788 | 25 |
| Shanghai | 0.984 | 10 | Inner Mongolia | 0.784 | 26 |
| Chongqing | 0.964 | 11 | Liaoning | 0.758 | 27 |
| Sichuan | 0.951 | 12 | Tianjin | 0.757 | 28 |
| Shaanxi | 0.941 | 13 | Ningxia | 0.744 | 29 |
| Shandong | 0.937 | 14 | Qinghai | 0.714 | 30 |
| Yunnan | 0.892 | 15 | Jilin | 0.694 | 31 |
| Guangxi | 0.882 | 16 | | | |

According to the risk level in Table 3, the risk level of provinces or provincial municipalities weighted by Baidu migration index is shown in Table 12.

From the information reflected in Tables 10–12, the epidemic index weighted by Baidu migration index can accurately reflect the actual epidemic information. The difference between the predicted epidemic ranks and the real one is relatively small. In one hand, 1 (Beijing) and 3 (Hebei) have also risen from the third level of risk cities predicted by geographical-economic distance to the second level of risk cities. In the other hand, 26 (Tibet), 28 (Gansu) and 31 (Xinjiang) are economically backward provinces. Based on the geographical-economic distance prediction, they are third-level dangerous provinces, and a well-corrected migration index weighting based on Baidu. In the prediction of provinces or provincial municipalities with a difficult epidemic situation, we note a reversed prediction ranking for Zhejiang and Jiangsu. Because Jiangsu's economic strength is higher than Zhejiang's, and after the outbreak, there were 18,800 people from Wuhan, Hubei Province to Wenzhou, Zhejiang Province from January 23 to 27

**Table 11 Rank difference between the epidemic index ranking and real epidemic ranking.**

| Region | Prediction rank | Real rank | Rank difference | Region | Prediction rank | Real rank | Rank difference |
|---|---|---|---|---|---|---|---|
| Hubei | 1 | 1 | 0 | Hebei | 17 | 15 | 2 |
| Hunan | 2 | 5 | 3 | Beijing | 18 | 13 | 5 |
| Henan | 3 | 3 | 0 | Hainan | 19 | 20 | 1 |
| Guangdong | 4 | 2 | 2 | Guizhou | 20 | 21 | 1 |
| Jiangxi | 5 | 7 | 2 | Gansu | 21 | 26 | 5 |
| Anhui | 6 | 6 | 0 | Shanxi | 22 | 23 | 1 |
| Jiangsu | 7 | 9 | 2 | Xinjiang | 23 | 27 | 4 |
| Zhejiang | 8 | 4 | 4 | Heilongjiang | 24 | 12 | 12 |
| Fujian | 9 | 16 | 7 | Tibet | 25 | 31 | 6 |
| Shanghai | 10 | 14 | 4 | Inner Mongolia | 26 | 28 | 2 |
| Chongqing | 11 | 10 | 1 | Liaoning | 27 | 24 | 3 |
| Sichuan | 12 | 11 | 1 | Tianjin | 28 | 22 | 6 |
| Shaanxi | 13 | 18 | 5 | Ningxia | 29 | 29 | 0 |
| Shandong | 14 | 8 | 6 | Qinghai | 30 | 30 | 0 |
| Yunnan | 15 | 19 | 4 | Jilin | 31 | 25 | 6 |
| Guangxi | 16 | 17 | 1 | Average Rank | – | – | 3.1 |

**Table 12 Risk classification of provinces or provincial municipalities based on epidemic index.**

| Hazard level | Division results of epidemic index | Real division results |
|---|---|---|
| Level one | 17 | 17 |
| Level two | 10, 12, 14, 16, 18, 19 | 11, 12, 14, 16, 18, 19 |
| Level three | 1, 3, 9, 11, 13, 15, 20, 21, 22, 23, 24, 25, 27 | 1, 3, 8, 9, 10, 13, 15, 20, 21, 22, 23, 25, 27 |
| Level four | 2, 4, 5, 6, 7, 8, 10, 26, 28, 29, 30, 31 | 2, 4, 5, 6, 7, 24, 26, 28, 29, 30, 31 |

(https://baijiahao.baidu.com/s?id=1657417825137559938&wfr=spider&for=pc), which led to the deviation of our model prediction.

The prediction ranking deviation of Heilongjiang Province is significant, because Harbin, as a famous ice and snow tourism city, is the first choice of many tourists, especially in the epidemic area, which provides objective conditions for epidemic input. Before the outbreak, there was a massive flow of people during the ice and snow tourism season. Data shows that from December 1, 2019, to January 31, 2020, despite the impact of the epidemic, Harbin still receives about 70,000 Hubei tourists, including 43,899 registered accommodation and 10,450 Wuhan tourists (http://www.hlj.chinanews.com/hljnews/2020/0210/55385.html).

## RESULTS

In the empirical "Epidemic Prediction Based on Spatial Geographic Adjacency Information", we observe that classifying the epidemic level only by relying on the geographic adjacency information (GAI) and the information of the epidemic origin have

| Table 13 Confusion matrix. | | | | |
|---|---|---|---|---|
| Level | 1 | 2 | 3 | 4 |
| (A) Confusion matrix based on GAI | | | | |
| 1 | 1 | 0 | 0 | 0 |
| 2 | 0 | 4 | 2 | 0 |
| 3 | 0 | 2 | 6 | 5 |
| 4 | 0 | 0 | 5 | 6 |
| Accuracy | 17/31 = 54.84% | | | |
| (B) Confusion matrix based on EID | | | | |
| 1 | 1 | 0 | 0 | 0 |
| 2 | 0 | 5 | 1 | 0 |
| 3 | 0 | 1 | 9 | 3 |
| 4 | 0 | 0 | 3 | 8 |
| Accuracy | 23/31 = 74.19% | | | |
| (C) Confusion matrix based on GED | | | | |
| 1 | 1 | 0 | 0 | 0 |
| 2 | 0 | 5 | 1 | 0 |
| 3 | 0 | 1 | 9 | 4 |
| 4 | 0 | 0 | 4 | 7 |
| Accuracy | 22/31 = 70.97% | | | |
| (D) Confusion matrix based on BMI | | | | |
| 1 | 1 | 0 | 0 | 0 |
| 2 | 0 | 5 | 1 | 0 |
| 3 | 0 | 1 | 11 | 1 |
| 4 | 0 | 0 | 1 | 10 |
| Accuracy | 27/31 = 87.10% | | | |

achieved an accuracy of 54.84% (Table 13A). This method is very important for policy control and deployment in the early stage of the outbreak. This is a way to deploy control measures as soon as an outbreak is detected, and it can respond quickly in a very short period.

In the empirical "Epidemic Prediction Based on Spatial Euclidean Inverse Distance", we added the information of distance, which is more reasonable than geographical adjacency information, because the farther the distance between the two places is, the less conducive it is to the spread of infectious diseases. The accuracy of this method is 74.97% (Table 13B). The classification of an epidemic situation based on Euclidean inverse distance (EID) information can also respond quickly in a short time.

We also include the information of economic distance in the empirical "Epidemic Prediction Based on Geographical Distance and Economic Distance", because the economic relationship will affect the spread of the epidemic. Although this method does not substantially improve the accuracy of epidemic classification, but make the ranking of epidemic index more reasonable (Table 13C).

Furthermore, we consider the information of the Baidu migration index (BMI) in the empirical "Epidemic Prediction Based on Migration Index Weighting of Baidu", to take into account the effect of the population flow in the spread of the epidemic. Based on the method of comprehensive information, we have improved accuracy of 87.10% (Table 13D). These methods do not use any information on the epidemic itself, which is significant for the guidance of the next outbreak.

The study discusses the epidemic situation based on geographic adjacency information (GAI), Euclidean inverse distance (EID), geographical-economic distance (GED) and Baidu migration index (BMI). Tables 13A–13D provides the predicted and real results. Confusion matrix (Table 13) shows that the accuracy of our prediction results has increased from 54.84% to 87.10%, and the prediction results are stable. The dynamic spread of the epidemic does not reduce the robustness of our model. Our paper was written in March 2020 and revised in July 2020, but the difference between the predicted level and the actual epidemic development remains unchanged.

## DISCUSSION

Many papers analyze the data set of COVID-19 (*Benvenuto et al., 2020*; *Chung, Ho & Wu, 2020*; *Yadav, 2020*; *Yang et al., 2020*), diagnose and treat the epidemic situation according to the information reflected in the data set, and give the corresponding control measures. This kind of achievements only has the summary guidance significance to the epidemic situation itself, but does not have the corresponding generalization ability. Because in the next outbreak, there is no relevant COVID-19 data set information, and the virus of the infectious diseases might change. In this paper, we try to avoid using the information of COVID-19 data set for epidemic simulation and risk assessment.

There are also some papers that predict the spread of the epidemic. They used the epidemic data to construct SIR model (*Comunian, Gaburro & Giudici, 2020*), real-time model (*Alberti & Faranda, 2020*), link network model (*Hazarika & Gupta, 2020*) and risk assessment model (*Koutsellis & Nikas, 2020*). These models used the data of epidemic situation itself, and they also achieved good prediction effect. The less data is needed for the prediction of epidemic spread and risk, the better. The best model is to issue corresponding control measures in a short time. Because time is the most important chip for the people or country concerned to win the battle against epidemic situation.

The research results of this article are of considerable practical significance, but there are still some areas to be improved in the future.

First, as the global epidemic continues to spread, imported cases should be considered when predicting the spread of the epidemic in Mainland China. One can add the import and export volume of each province to simulate the spread of the epidemic.

Second, the transmission speed of the novel coronavirus to various provinces and municipalities in Mainland China is also related to the traffic level of various provinces and municipalities, hence, can be simulated by the number of railways and highways in various provinces and municipalities.
Third, the epidemic prevention and control measures of a province have a tangible impact on the spread of the epidemic. We can position the quality of epidemic prevention and control measures through the data of public opinion.

Although the research approach of this manuscript needs to be improved to achieve more comprehensive and systematic results. But it does achieve a good prediction accuracy, and the accuracy of this prediction does not appear any deviation with the development of the epidemic, so the robustness of the model is very good. Therefore, for researchers and policy makers, the policies and measures in next outbreak can be based on the results of our model. When the first case is found, we can quickly classify the epidemic level in China in the future. According to the level results, the corresponding level of medical and health response was carried out. This is the optimal strategy between total closure and free control, which can minimize the economic loss and case infection rate.

## CONCLUSIONS

In this article, we simulate the spread of novel coronavirus disease and used geographical adjacency information, Euclidean spatial distance, geographical-economic distance, and Baidu migration index to predict the spread of the epidemic index and risk level of each province. The conclusions of this paper are as follows.

First, the accuracy of forecasting the risk level of provinces or provincial municipalities based on merely geographical adjacency information is about 54.84%. There are some differences between the simulation results and the actual epidemic situation, and the prediction results have specific reference value. This is because, in the early stage of epidemic development, we do not know much about the virus and the spatial information of the whole region. In a short period, we can start the corresponding level response of major public health emergencies according to simple geographic information.

Secondly, the accuracy of forecasting the risk level of provinces or provincial municipalities based on the spatial Euclidean inverse distance and the geographical-economic distance matrix is about 70%. However, it indicates a better ranking of the epidemic index based on the geographical-economic matrix is more reasonable than that of the spatial Euclidean inverse distance. The simulated results reveal that the spread of the epidemic is related to the economic level of each province. That is, the higher the economic level of each province, the closer the economic exchanges with other provinces, which provides objective conditions for the large-scale spread of the epidemic. Therefore, when provinces or provincial municipalities start to respond to the level of major public health emergencies, the provinces or provincial municipalities with the highest economic level can appropriately improve the level based on the neighboring relationship.

Thirdly, based on the geographical-economic distance matrix and Baidu migration index, the accuracy rate of the epidemic risk level prediction is 87.10%, which can reflect the real epidemic situation. Through the simulation of these non-epidemic data, we can get the ranking of the epidemic index and the risk level of each province, which has an outstanding practical value.

According to the predicted results, the provincial and municipal governments and medical and health institutions can start the corresponding response of major public health emergencies, and make preparation for medical and health care, and issue a control measures to prevent the spread of the epidemic. These can minimize economic loss and the number of infected people from the epidemic.

The research results of this article can not only be used as a reference for national prevention and control measures, but also as a reference sample for the prevention and control measures of each prefecture-level city within a province, and even as a vital basis source for the prevention and control measures of other countries in the world. Government officials, experts and scholars of other countries can simulate the spread of epidemic according to their country's situation, predict the spread of epidemic index and the corresponding risk level of provinces or provincial municipalities. Going by this can reduce the risk of the epidemic spread and national economic loss.

## ACKNOWLEDGEMENTS

The authors want to thank all reviewers for their valuable and constructive comments, which significantly improves the quality of this paper.

### Funding
Xuewei Cheng was supported by the Postgraduate innovation project in Hunan Province (No. CX20200148). The funders had no role in study design, data collection and analysis, decision to publish, or preparation of the manuscript.

### Grant Disclosures
The following grant information was disclosed by the authors:
Postgraduate innovation project in Hunan Province: CX20200148.

### Competing Interests
The authors declare that they have no competing interests.

### Author Contributions

- Xuewei Cheng conceived and designed the experiments, performed the experiments, analyzed the data, prepared figures and/or tables, authored or reviewed drafts of the paper, and approved the final draft.
- Zhaozhou Han conceived and designed the experiments, analyzed the data, prepared figures and/or tables, authored or reviewed drafts of the paper, and approved the final draft.
- Badamasi Abba analyzed the data, prepared figures and/or tables, authored or reviewed drafts of the paper, and approved the final draft.
- Hong Wang conceived and designed the experiments, performed the experiments, analyzed the data, prepared figures and/or tables, authored or reviewed drafts of the paper, and approved the final draft.

## Data Availability

The data is available in the Supplemental Files.

## Supplemental Information

Supplemental information for this article can be found online at http://dx.doi.org/10.7717/peerj.10139#supplemental-information.

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
