# Peer review of "Regional infectious risk prediction of COVID-19 based on geo-spatial data"

_PeerJ, doi:10.7717/peerj.10139_

## Round 0.1 · original submission · Major Revisions

Three experts in the field have reviewed your manuscript. Based on their comments, the manuscript has been returned for major revision, giving that the study is new and interesting, but it requires substantial work to improve its data presentation and value, such as a clear rationale of the study, data update, figure(s) for better understanding the results, and extensive literature review and appropriate citations. If you decide to revise the manuscript, please address the reviewers' concerns (listed below) point-by-point.

·

Basic reporting

The manuscript subject is new and interesting but it should be corrected as follow comments;
1- The introduction is well-written however it should concentrate on the novelty and importance of the work.
2- Equation should be enumerated and cited in the manuscript.
3- The format of commas is not correct.
4- the manuscript is the lock of figures. it needs some figures or plot for reader to be helpful.
5- The manuscript should consider the sensitivity analysis to find the effective parameters. it should also add to the literature review, Investigation of effective climatology parameters on COVID-19 outbreak in Iran(Science of the Total Environment), Environmental Science and Pollution Research (Modeling and sensitivity analysis of NOx emissions and mechanical efficiency for diesel engine), Application of gene expression programming and sensitivity analyses in analyzing effective parameters in gastric cancer tumor size and location(Soft Computing),
6- You should also use some artificial intelligence methods for analysis. you should also use prediction methods of

Presentation of new thermal conductivity expression for –water and CuO–water nanofluids using gene expression programming (Journal of Thermal Analysis and Calorimetry), Presentation of a new hybrid approach for forecasting economic growth using artificial intelligence approaches(Neural Computing and Applications0, A gene expression programming model for economic growth using knowledge-based economy indicators(Journal of Modelling in Management), Statistical and Econometrical Analysis of Knowledge-Based Economy Indicators Affecting Economic Growth in Iran: The new evidence of Principal Component Analysis-Tukey and ARDL(Journal of Policy Modeling).

Experimental design

the manuscript has an Experimental data and should explain the data description statistic

Validity of the findings

validation the manuscript is incorrect and should be improved

Reviewer 2 ·

Basic reporting

The English is mostly clear but it could do with some editing.

The major issue with this research, and it is for this reason that I recommend that it is rejected, is that there is an absence of references to previous works. There need to be references cited backing up the method used by the researchers. I realize that this is a new pandemic but previous epidemiological studies can be referenced as can the motivation for the research methodology used. This is one of the basic principals of scientific research and as such it should not be published in its current form.

It is very strange that this is a paper on geospatial methods and yet no maps or Figures of any description are provided.

The paper refers to a submission date of March and yet I only received this paper more than three months later suggesting that this is not the first journal to receive this paper. Especially given that this is a fastly moving pandemic, the data should be as up to date as possible.

Experimental design

No comment

Validity of the findings

No comment

Additional comments

It may well be that the paper has a contribution to add to the understanding of the pandemic. Unfortuantely it reads more like a report than a scientific paper and therefore I have to recommend that it is rejected. If the authors wish to try and publish elsewhere, I recommend that they make sure the pandemic information is as up to date as possible and they follow the generally accepted approach of making sure the research is placed within the established body of litereature.

Reviewer 3 ·

Basic reporting

An extended background on Covid-19 is covered in Introduction. Introduction should be written clearly and concisely on the problem the authors would like to address and why it is an important problem to tackle. The Introduction section should also contains the description of the approach along with the brief discussion on the justification to support the approach taken. The findings of the approach as well as the contributions of the paper are not highlighted. The approach employed by the authors has practical value that merits sharing with the wider academic community, but more work needs to be done to emphasize their technical aspects as well as to more convincingly demonstrate the original contribution that this work makes to the scholarly knowledge base. Both findings and contributions should be clearly stated in the introduction part of the paper.

A vague and inconsistent expression such as “more than two months ago”, “geographic information adjacency” instead of “geographic adjacency information” throughout the paper should be avoided. In terms of literature review, a list of COVID related studies and a brief literature review on risk in coronavirus studies are described in the Introduction section. An important first step that will need to be taken to address this shortcoming is the inclusion of a more comprehensive literature review. In its present form, the brief discussion of existing work in the Introduction is inadequate in breadth and depth. While cursory mention is made in this section of other studies and the fact that they do not support risk mapping inside Mainland China, there is no attempt to present a comparison of their features, strengths, and weaknesses, and more importantly, to show how the present work builds upon and extends what has already been done. Also conspicuously missing from the literature review is coverage of the relevant scientific literature on the use of spatial indexes, economic indexes, and mobility indexes and the intersection between these areas, especially as they relate to surveillance and prediction of risk within both infectious and non-infectious diseases.

Furthermore, additional visualizations such as the map of Mainland China with provincial information as well as the relevant index numbers from (Table 1) will increase the readability for the readers who are not familiar with the topology of the provinces. Some tables need clarification particularly on the use of different colors (red vs. blue) of fonts. Instead of varying the colors, varying the font style such as Bold vs. Italics is recommended.

Experimental design

The "Methods and Results" section gives a step by step description of the method. This part of the manuscript could be enhanced through the addition of more detailed technical on “programming and simulation” in every step of the method as well as justification of the decisions and choices made, with reference to extant research findings and (spatial risk mapping) theory that underpinned or informed the various aspects of the study approach. It would also be worthwhile outlining alternative techniques and approaches that may have been considered, including the reasons they were deemed unsuitable.

Validity of the findings

As part of the evaluation, the results are evaluated against the actual risk division results from a news website on daily reporting information about the disease. Coronavirus is an ongoing pandemic and this study compare the results with the information from one specific point in time, it is important to report and discuss the source, and when the snapshot was taken to further validate the results.
The "Discussion" section is needed to add. If the manuscript is to move beyond mere "show and tell" to make a real contribution to the field, the Discussion will need to be substantially developed to (a) situate the outcomes, findings, and lessons learned from the study in the wider context of the literature surveyed in the (expanded) literature review; (b) offer evidence-based practical recommendations for researchers, policy planners and system developers that are grounded in the researchers’ findings and experiences/observations; and (c) consider how the present work might apply to or otherwise have value that transcends the discipline of risk mapping in epidemiology.

Additional comments

Line (52 - 67), existing work is described. More technical insights are needed and how and why the existing work is not suitable for the problem the authors are tackling?
Line (75 - 79) In the last paragraph of the introduction, the approach should be summarized and the findings should be briefly delivered? Importantly, the main contributions of the paper should be highlighted.
Line 82 -” more than two months ago”, needs clarification
Introduction and Inspiration sections can be combined together and add a related work section covering the discussion on how spatial proximity information, economic indexes, and mobility indexes have been used disease risk mapping in general and in particular coronavirus.

Line 97, 259, 263 - information adjacency -> adjacency information
Line 107 - “* the numbers of cities in this paper are in the order of Table 1.”, Table 1 is the list of provinces or provincial municipalities instead of cities.
Line 109 - “Based on the above geographical adjacency information, we partitioned the 31 provinces or provincial municipalities into 4 risk levels with Hubei as the center of infectious diseases.” Justification is needed for four risk levels and explanation on how the provinces are partitioned explicitly.
Line 109 - Table 2 presents instead of Table 2 present
Line 112 - table 2 -> Table 2
Line 122 - Hubei was stated twice.
Line 124 - add scholarly references to such Principle
Line 127 - clarify “simple programming and simulation”.
Line 132 - full stop (not a comma) after Table 2.
Line 138 - The sentence begins with In comparison with the real data, …..State the location of the real data in the paper? If it is Table 4, state it clearly.
Line 139 - 142: The sentence starts with “From the epidemic index: …”, to which table the authors are referring to? The indexes the authors referred to the sentence are the indexes in Table 1 which are not related to the epidemic index.
Line 146 - need a reference to “the real level division,”
Line 149 - need a reference to “the real situation”,
Line 157 - The sentence “Instead of geographical adjacency, with all of them given the same weight index.” Need to clarify “them” in the second part of the sentence.
Line 161 - It is express -> It is expressed
Line 166 - need a reference to theory of distance inverse weight
Line 172 - The sentence starts with “After programming and simulation, …”. Need to elaborate on the simulation, what was simulated and how it has been done?
Line 197 - In paragraph reference to Lin Guangping (2015) is not matched with the reference #26 (which is stated as 2005)
Line 228 - the same comment as Line 172.
Line 234 - table 10 -> Table 10.
Line 242 - The sentence starts with “In other hand, …”. The following sentence in Line 243 also starts with “In other hand,...”.
Line 244 - Need clarification on the phrase “three-level dangerous provinces”.
Line 259 - The spread novel coronavirus -> The spread of novel coronavirus
Line 259 - the same suggestion as at Line 172.
Line 259 - The sentence stated Euclidean spatial distance, geographical economic distance and Baidu migration index, and …….are predicted.”
“Euclidean spatial distance, geographical economic distance and Baidu migration index” are not predicted, they are used in predicting the risk of the regions in the study.
Line 263 - the authors stated that the accuracy of 54% for geographical information adjacency. Providing information such as (the ranks of xx/yy provinces are correctly predicted) will assist the reader.
Line 263 - geographical information adjacency -> geographical adjacency information
Line 264 - The sentence is not clear “there are some differences between the simulation results and the actual epidemic situation, and the prediction results have specific reference values”. What are the specific reference values? With 54% of accuracy, it is obvious that there are differences but need to state them stately.
Line 269 - the same suggestion as at Line 263
Tables 4, 6, 8, 11, what is the meaning of the use of different colors (blue and red)
Tables 4, 6, 8, when the real division results, when the information was accessed?

---

## Round 0.2 · Major Revisions

The same reviewers have commented on your revised manuscript, as listed below. Although the manuscript has been improved, some issues raised by the reviewers in the first round of the review and some other major concerns raised now need to be addressed.

I noticed that the revised manuscript has a very short "RESULTS' part, even shorter than the "CONCLUSION" part. I suggest that the authors revise the results section, and perhaps subdivide this section into topics with links to figures and tables.

To facilitate the review process, please show track changes in the revised manuscript, so the reviewers can easily follow what the authors have addressed their concerns.

·

Basic reporting

The manuscript has not been revised correctly. the comments need to be considered again as follows:

1- All of the reviewer's comments should be highlighted.
2- The references mentioned in the text are not in ref. for example, the Ahmadi et al., (2020) is not in ref.
3- The result of the manuscript is poor and the authors should revise again.
4- The authors should add the new section as a literature review:
- Presentation of a new hybrid approach for forecasting economic growth using artificial intelligence approaches (Neural Computing and Applications)
- Diagnosis and Detection of Infected Tissue of COVID-19 Patients Based on Lung X-Ray Image Using Convolutional Neural Network Approaches(Chaos, Solitons & Fractals)
- Application of gene expression programming and sensitivity analyses in analyzing effective parameters in gastric cancer tumor size and location(Soft Computing )
5- The discussion is very poor and the authors should mention the importance of their manuscript in comparison with other papers.

Experimental design

The manuscript has not been revised correctly. the comments need to be considered again as follows:

1- All of the reviewer's comments should be highlighted.
2- The references mentioned in the text are not in ref. for example, the Ahmadi et al., (2020) is not in ref.
3- The result of the manuscript is poor and the authors should revise again.
4- The authors should add the new section as a literature review:
- Presentation of a new hybrid approach for forecasting economic growth using artificial intelligence approaches (Neural Computing and Applications)
- Diagnosis and Detection of Infected Tissue of COVID-19 Patients Based on Lung X-Ray Image Using Convolutional Neural Network Approaches(Chaos, Solitons & Fractals)
- Application of gene expression programming and sensitivity analyses in analyzing effective parameters in gastric cancer tumor size and location(Soft Computing )
5- The discussion is very poor and the authors should mention the importance of their manuscript in comparison with other papers.

Validity of the findings

The manuscript has not been revised correctly. the comments need to be considered again as follows:

1- All of the reviewer's comments should be highlighted.
2- The references mentioned in the text are not in ref. for example, the Ahmadi et al., (2020) is not in ref.
3- The result of the manuscript is poor and the authors should revise again.
4- The authors should add the new section as a literature review:
- Presentation of a new hybrid approach for forecasting economic growth using artificial intelligence approaches (Neural Computing and Applications)
- Diagnosis and Detection of Infected Tissue of COVID-19 Patients Based on Lung X-Ray Image Using Convolutional Neural Network Approaches(Chaos, Solitons & Fractals)
- Application of gene expression programming and sensitivity analyses in analyzing effective parameters in gastric cancer tumor size and location(Soft Computing )
5- The discussion is very poor and the authors should mention the importance of their manuscript in comparison with other papers.

Additional comments

The manuscript has not been revised correctly. the comments need to be considered again as follows:

1- All of the reviewer's comments should be highlighted.
2- The references mentioned in the text are not in ref. for example, the Ahmadi et al., (2020) is not in ref.
3- The result of the manuscript is poor and the authors should revise again.
4- The authors should add the new section as a literature review:
- Presentation of a new hybrid approach for forecasting economic growth using artificial intelligence approaches (Neural Computing and Applications)
- Diagnosis and Detection of Infected Tissue of COVID-19 Patients Based on Lung X-Ray Image Using Convolutional Neural Network Approaches(Chaos, Solitons & Fractals)
- Application of gene expression programming and sensitivity analyses in analyzing effective parameters in gastric cancer tumor size and location(Soft Computing )
5- The discussion is very poor and the authors should mention the importance of their manuscript in comparison with other papers.

Reviewer 3 ·

Basic reporting

-

Experimental design

-

Validity of the findings

-

Additional comments

The manuscript is substantially improved but the English language needs to improve further. The use of zero epidemic data is relatively common in predicting the risk of a disease or disease outbreak in epidemiology. The authors have added and cited literature works in the Introduction, however, the related work surveyed needed to be connected together and discussed in an organized and concise manner. The background on the COVID-19 timeline in mainland China in Introduction should be succinctly described with the essential facts to highlight the problem the paper would like to address only. The synthesized information on the problem described in the introduction, the findings from the current study with the supporting literature, should be wrapped up in both the discussion and conclusion sections.

---

## Round 0.3 · accepted · Accept

The authors have addressed the reviewers' concerns, and the manuscript has now been accepted for publication.

·

Basic reporting

The revised has been correctly and the manuscript is suitable for publication

Experimental design

it is good

Validity of the findings

it is good

Additional comments

the revised has been correctly and the manuscript is suitable for publication